# Synthesis of Bimetallic Ni-Co Phosphide Nanosheets for Electrochemical Non-Enzymatic H_2_O_2_ Sensing

**DOI:** 10.3390/nano13010066

**Published:** 2022-12-23

**Authors:** Zhi-Yuan Wang, Han-Wei Chang, Yu-Chen Tsai

**Affiliations:** 1Department of Chemical Engineering, National Chung Hsing University, Taichung 40227, Taiwan; 2Department of Chemical Engineering, National United University, Miaoli 360302, Taiwan; 3Pesticide Analysis Center, National United University, Miaoli 360302, Taiwan

**Keywords:** NiCoP nanosheets, phosphorization, non-enzymatic detection of H_2_O_2_, electrocatalytic active sites

## Abstract

NiCoP nanosheets (NSs) were successfully synthesized using the hydrothermal and high-temperature phosphorization process. The obtained NiCoP NSs were immobilized on a glassy carbon electrode (GCE) and used to construct a novel sensing platform for electrochemical non-enzymatic H_2_O_2_ sensing. Physicochemical characteristics of NiCoP NSs were obtained by field-emission scanning electron microscopy (FESEM), field-emission transmission electron microscope (FETEM), X-ray diffraction (XRD), and X-ray photoelectron spectroscopy (XPS). In addition, the electrochemical properties of NiCoP NSs were obtained by cyclic voltammetry (CV) and chronoamperometry (CA) towards the non-enzymatic detection of H_2_O_2_. FESEM and FETEM images provided a morphological insight (the unique nanosheets morphology of NiCoP) that could expose more active sites to promote mass/charge transport at the electrode/electrolyte interface. XRD and XPS results also confirmed the crystalline nature of the NiCoP nanosheets and the coexistence of multiple transitional metal oxidation states in NiCoP nanosheets. These unique physicochemical characteristics had a degree of contribution to ensuring enhancement in the electrochemical behavior. As a result, the synthesized NiCoP NSs composed of intercalated nanosheets, as well as the synergistic interaction between bimetallic Ni/Co and P atoms exhibited excellent electrocatalytical activity towards H_2_O_2_ electroreduction at neutral medium. As the results showed, the electrochemical sensing based on NiCoP NSs displayed a linear range of 0.05~4 mM, a sensitivity of 225.7 μA mM^−1^ cm^−2^, a limit of detection (LOD) of 1.190 μM, and good selectivity. It was concluded that NiCoP NSs-based electrochemical sensing might open new opportunities for future construction of H_2_O_2_ sensing platforms.

## 1. Introduction

As a reactive oxygen species (ROS), hydrogen peroxide (H_2_O_2_) serves as a typical biomarker for a lethal tumor microenvironment, which is implicated in attenuating anti-tumoral immune response and promoting tumor growth in various cancers. It indicates that H_2_O_2_ plays a key role in tracking cancer development and progression. Mild levels of H_2_O_2_ act as a second messenger involved in signaling controlling cell proliferation, differentiation, and survival, and gene transcription regulation. The evidence presents that the occurrence of the transformation of normal cells into tumor cells (carcinogens) induces an increase in H_2_O_2_ production, which causes a disturbance in the intracellular redox balance to trigger oxidative stress, thereby further leading to the initiation of cell death, tissue damage, and even cardiovascular diseases and neuronal degeneration [1]. Therefore, it is highly desirable to develop effective and accurate technology monitoring the level of H_2_O_2_, which can be regarded as a biological indicator to reflect the development and progression of cancer cells.

To date, a variety of analytical methods have been developed to detect H_2_O_2_ accurately, including fluorescence [2], chemiluminescence [3], chromatography [4], and electrochemistry [5,6,7]. Among these conventional analytical methods, the detection of target molecules using electrochemical methods is widely used because of their advantages of low cost, easy portability, high sensitivity, and fast response [8,9]. According to the present research progress, electrochemical methods for the detection of target molecules could be further subdivided into electrochemical enzymatic and non-enzymatic sensing. In comparison to electrochemical non-enzymatic sensing, natural enzymes in the construction of enzymatic sensing are normally insufficiently sensitive, time-consuming, expensive, and environment-susceptible, limiting their application given a decrease in sensing performance [10]. Electrochemical non-enzymatic sensing is therefore developed to overcome these limitations.

It is well-known that nanomaterials with extraordinary properties are key factors in the fabrication of electrochemical non-enzymatic sensing. Noble metals and transition-metal compounds have been explored for their potential in the development of electrochemical non-enzymatic sensing. Transition metal compounds (TMCs) including transition-metal oxides (TMOs) [11], sulfides (TMSs) [12], nitrides (TMNs) [13], and phosphides (TMPs) [14,15] can deliver the benefits of more easily controllable synthesis, higher activity, and lower cost, which have been considered as ideal electrode materials and strong contenders to be integrated into electrochemical sensing. Among transition-metal compounds, transition-metal phosphides (TMPs) possess metalloid characteristics, excellent electrocatalytic activity, and good electrical conductivity [16,17]. Meanwhile, the relatively low electronegativity of P (2.19) compared with S (2.58), N (3.04), and O (3.44), leads to an increase in the covalency of metal−P bonding. This means that the higher energy and weaker attraction of electrons in the 3d orbitals of transition metal atoms, through relatively strong covalent metal−ligand interactions, promotes excellent reaction kinetics to improve electrochemical performance. Recent advances on transition metals, such as nickel (Ni) and cobalt (Co), possess advantages in electrochemical sensing due to their high content on the earth, low cost, low toxicity, and high availability. Furthermore, the Ni and Co elements located in the first row of transition metals have unique electrocatalytic effects due to their distinct d-electron structure [18]. Compared with single metallic Ni or Co transition-metal compounds, bimetallic Ni–Co transition-metal compounds, owing to a positive synergistic effect between Ni and Co, have received extensive interest and have been demonstrated to boost electrocatalytic performance in electrochemical sensing applications [19].

In this work, NiCoP nanosheets (NSs) were successfully synthesized using the hydrothermal and high-temperature phosphorization process. NiCoP NSs with unique characteristics offered high accessibility of active sites and rapid carriers transfer, and further the positive synergistic effects of Ni–Co sites and P sites within NiCoP NSs. This was expected to improve the electrocatalytic activity for H_2_O_2_ sensing.

## 2. Materials and Methods

### 2.1. Reagents

Cobalt (II) sulfate heptahydrate (CoSO_4_·7H_2_O), sodium phosphate dibasic (Na_2_HPO_4_), sodium phosphate monobasic (NaH_2_PO_4_), nickel (II) sulfate hexahydrate (NiSO_4_·6H_2_O), and sodium hypophosphite monohydrate (NaH_2_PO_2_·H_2_O) were purchased from Alfa Aesar (Ward Hill, MA, USA). Glycerol and anhydrous ethanol (C_2_H_5_OH, 99.9%) were purchased from J.T. Baker (Phillipsburg, NJ, USA). Hydrogen peroxide (H_2_O_2_), urea, Nafion solution (5 wt% in mixture of lower aliphatic alcohols and water), uric acid (UA), L-Ascorbic acid (AA), dopamine hydrochloride (DA), D-(+)-Glucose (Glu), and sodium chloride (NaCl) were purchased from Sigma-Aldrich (St. Louis, MO, USA). The deionized water (DI water) was produced from a Milli-Q water purification system (Millipore, Milford, MA, USA). All chemicals were analytical grade and were used as received without further purification.

### 2.2. Synthesis of NiCoP Nanosheets (NSs)

The NiCoP nanosheets (denoted as NiCoP NSs) were synthesized by a facile hydrothermal method, followed by a phosphorization process. 840 g CoSO_4_·7H_2_O, 0.394 g NiSO_4_·6H_2_O, and 0.15 g urea were mixed in a solution containing DI water and glycerol (V_DI water_:V_glycerol_ = 5:1). After stirring the mixture for 20 min, the mixture was transferred into a 100 mL Teflon-lined stainless autoclave and heated to 170 °C for 20 h in an oven. The resulting precipitate was collected and washed with DI water and ethanol 3 times by centrifugation, and then dried in an oven at 70 °C overnight. 

The obtained NiCo precursor (denoted as NiCo-Pre) and NaH_2_PO_2_·H_2_O in a mass ratio of 2:5 (for comparison, another mass ratio of 1:10 was also used) were placed in two separated alumina boats. The boat containing NaH_2_PO_2_·H_2_O was put on the upstream side of the tube furnace and another boat containing NiCo precursor was put on the downstream side of the tube furnace. Subsequently, the NiCo precursor was calcined at 300 °C for 2 h with a heating rate of 2 °C min^−1^ (for comparison, a heating rate of 0.5 and 5 °C min^−1^ was also used) under argon atmosphere, and then naturally cooled to room temperature. For comparison, NiP without a Co source and CoP without a Ni source were also fabricated by a similar route. The reaction mechanism of NiCoP formation is similar to the previous report [20] and can be expressed as the following Reaction Equations (1) and (2):2 NaH_2_PO_2_ → Na_2_HPO_4_ + PH_3_ ↑(1)
2 PH_3_ + 2 NiCo → 2 NiCoP + 3 H_2_ ↑(2)

### 2.3. Fabrication of NiCoP NSs Modified Glassy Carbon Electrode

To clean the glassy carbon electrode (GCE, diameter 3 mm, Tokai Carbon, Tokyo, Japan), the bare GCE was carefully polished with 0.3 and 0.05 μm alumina slurries, respectively, and cleaned by brief ultrasonic treatment with DI water and ethanol, then dried in a 70 °C oven. A total of 2 mg of NiCoP NSs was dispersed in 1 mL of 0.5 wt% Nafion solution by ultrasonic for 30 min to form a uniform suspension. To prepare the modified electrode, 6 μL of NiCoP NSs suspension was cast on the precleaned GCE and dried in a 70 °C oven as the working electrode. Finally, the NiCoP/Nafion/GCE (denoted as NiCoP/GCE) was obtained and NiP/Nafion/GCE (denoted as NiP/GCE) and CoP/Nafion/GCE (denoted as CoP/GCE) were also prepared for comparison in the same route for the following electrochemical measurement.

### 2.4. Characterizations

The morphology was characterized using field-emission scanning electron microscopy (FESEM, JSM-7410F, JEOL, Akishima, Japan) and field-emission transmission electron microscopy (FETEM, JEM-2100F, JEOL, Akishima, Japan). The chemical structure and composition were determined by X-ray photoelectron spectroscopy (XPS, PHI-5000 Versaprobe, ULVAC-PHI, Chigasaki, Japan). The crystal phase was characterized using X-ray diffraction (XRD) (D8 Discover X-ray diffractometer with Cu Kα radiation (Bruker, Karlsruhe, Germany)). Electrochemical measurements were performed using a three-electrode system composed of as-prepared samples of modified GCE, a platinum wire counter electrode, and an Ag/AgCl (3 M KCl) reference electrode by an electrochemical analyzer (Autolab, model PGSTAT30, Eco Chemie, Utrecht, The Netherlands). All electrochemical measurements were conducted in 0.1 M phosphate-buffered saline (PBS, pH = 7.0) as the supporting electrolyte in the absence and presence of H_2_O_2_ at ambient temperature. Cyclic voltammetry (CV) curves were performed between 0~−0.8 V. Chronoamperometry was carried out at −0.55 V under magnetic stirring.

## 3. Results and Discussion

The structure and phase composition of NiCoP, NiP, and CoP were characterized by an XRD pattern, as shown in Figure 1. Compared with the standard diffraction pattern of hexagonal NiCoP (JCPDS No. 71-2336) [21,22], the XRD pattern of NiCoP shows six distinct diffraction peaks located at 2θ about 41.0°, 44.9°, 47.6°, 54.4°, 54.7°, and 55.3°, which can be attributed to the (111), (201), (210), (300), (002), and (210) planes. This result verifies the two-dimensional structure of NiCoP. In addition, the XRD pattern of NiP and CoP matches the standard diffraction pattern (JCPDS No. 74-1385) and the standard diffraction pattern (JCPDS No. 29-0497) [21,22], respectively.

The morphologies of NiCoP, NiP, and CoP were characterized by FESEM (Figure 2a–c) and FETEM (Figure 2d–f). It can be observed that NiCoP and NiP are composed of nanosheets with a size of about 300~400 nm. Interestingly, two-dimensional morphology can be discovered in transition-metal compounds [23], metal oxides [24], and carbon materials [25], which provide a morphological insight that could expose more active sites to promote mass/charge transport at the electrode/electrolyte interface as well as improving electrochemical performance. From the morphology of FESEM images, it is found that the intercalated nanosheets have many pores and a large specific surface area, providing more exposed catalytic active sites. In addition, comparing the FETEM images of NiCoP and NiP, it is found that NiCoP nanosheets have greater specific surface area and rougher surface than that of NiP, revealing attractive structure/surface morphological characterization. It is known that electrocatalytic activity is greatly dependent on the electrochemical active sites. The generated NiCoP nanosheets with large surface area lead to more active sites being exposed in the electrochemical experiments, which facilitate mass transport and fast charge transfer, resulting in improved electrochemical performance. From Figure 2c,f, it can be observed that CoP is the coexistence of nanosheets and filaments. Scanning transmission electron microscopy (STEM) and corresponding elemental Ni, Co, and P mapping images (Figure 2g–j) further demonstrate that Ni, Co, and P elements are uniformly distributed throughout NiCoP nanosheets.

The surface elemental composition and valance states of NiCoP, NiP, and CoP were characterized by XPS, as shown in Figure 3. Figure 3a shows the high-resolution Ni 2p XPS spectra of NiCoP and NiP. The Ni 2p XPS spectra are divided into two spin–orbit doublets (Ni 2p_1/2_ and Ni 2p_3/2_). To identify the specific Ni species, the Ni 2p_1/2_ (Ni 2p_3/2_) doublets are further deconvoluted into three peaks located at about 873.8 eV (856.1 eV) and 879.6 eV (860.9 eV), attributed to the Ni^2+^ and shake-up satellite (Sat.). Other peak features located at 869.9 (Ni 2p_1/2_) and 852.7 eV (Ni 2p_3/2_) corresponding to the Ni^δ+^ were found (more positively shifted compared with metallic Ni). The peak intensity of Ni^δ+^ peak for NiCoP is slightly higher than that of NiP, suggesting a higher valence state of Ni for NiCoP [26,27]. Figure 3b shows the high-resolution Co 2p XPS spectra of NiCoP and CoP. The Co 2p XPS spectra included two spin–orbit-split doublets (Co 2p_1/2_ and Co 2p_3/2_) located at about 797.4 and 781.5 eV, and two shake-up satellites (Sat.) located at about 802.6 (Co 2p_1/2_) and 785.5 (Co 2p_3/2_) eV, demonstrating the presence of a Co^2+^ valence state for NiCoP. Additionally, the Co 2p XPS spectra of CoP reveal clear differences in the characteristic peaks located at 793.9 (Co 2p_1/2_) and 779.1 (Co 2p_3/2_) eV corresponding to the Co^δ+^ (more positively shifted compared with metallic Co), suggesting a lower valence state of Co in NiCoP [28,29]. Figure 3c shows the high-resolution P 2p XPS spectra of NiCoP, NiP, and CoP. The P 2p XPS spectra were divided into three peaks. The peaks at about 130.6 (P 2p_1/2_) and 129.8 (P 2p_3/2_) eV corresponding to P^δ−^ in transition-metal phosphides and peak at about 133.4 eV corresponding to the PO_x_ can be attributed to the presence of oxidized P species. Compared with NiP and CoP, the binding energy of P^δ−^ in NiCoP has a slight shift toward higher binding energies, suggesting improved P 2p electron unoccupation in NiCoP [30,31]. The above XPS results further demonstrate the changes in electronic structures and support the existence of charge transfer between Ni/Co and P atoms in transition-metal phosphides during the phosphorization process, and confirms that the presence of the strong interaction between the transition metals (Ni/Co) and the phosphide atoms affords new opportunities for constructing transition-metal phosphides-based electrocatalysts, and further opens a promising route for boosting the electrocatalytic activities in electrochemical sensing.

The electrochemical characteristics of NiCoP, NiP, and CoP nanocomposites for H_2_O_2_ sensing were performed by cyclic voltammetry (CV). Figure 4 shows the CV curves of NiCoP/GCE, NiP/GCE, and CoP/GCE in 0.1 M PBS (pH 7.0) in the absence and presence of 5 mM H_2_O_2_ at a scan rate of 50 mV s^−1^ within the potential range of 0~−0.8 V. In the absence of H_2_O_2_, all of the NiCoP/GCE, NiP/GCE, and CoP/GCE display no obvious oxidation and reduction peaks of H_2_O_2_. Upon the addition of 5 mM H_2_O_2_, the reduction peak can be obviously found. In particular, NiCoP/GCE exhibits the maximum electrocatalytic reduction peak current at about −0.5 V for H_2_O_2_. These results indicate that NiCoP NSs have a better electrocatalytic effect on H_2_O_2_ electroreduction than that of NiP and CoP, most likely due to the synergistic interaction between bimetallic Ni/Co and P atoms, and their structure/surface morphological characterization. According to the previous report [32], transition-metal compounds as electrocatalysts might expose the catalytically active sites to further electrocatalyze the electrochemical reduction of H_2_O_2_. The reaction mechanism of H_2_O_2_ electroreduction in PBS in the presence of transition-metal compounds could be expressed as Equations (3)–(5), as shown below.
H_2_O_2_ + e^−^ → OH_ad_ + OH^−^(3)
OH_ad_ + e^−^ → OH^−^(4)
2OH^−^ + 2H^+^ → 2H_2_O(5)

Some synthetic parameters during the phosphorization process should be optimized in order to obtain the high performance of NiCoP NSs electrode materials for H_2_O_2_ electroreduction. As a control, the mass ratio of NiCo-Pre/NaH_2_PO_2_·H_2_O and the heating rate during the phosphorization process were investigated to gain better insight into the optimization of the operating parameters affecting electrochemical sensing performance. Figure 5a shows the CV curves of NiCoP synthesized with the different mass ratios of 1:10 and 2:5 (denoted as NiCoP(1:10) and NiCoP(2:5)) in the absence and presence of 5 mM H_2_O_2_ in 0.1 M PBS (pH 7.0). It can be observed from the results that NiCoP(2:5) exhibits the maximum electrocatalytic reduction peak current at the mass ratio of 2:5. It is speculated that the phosphorus (P) content in the NiCoP with the mass ratio of 1:10 is too high, which leads to excessive etching of catalytic metals Ni and Co by phosphorus gas (PH_3_) at high temperature, and then decreases the catalytic active sites. In Figure 5b, the CV curves of NiCoP are carried out at the different heating rates of 0.5 (denoted as NiCoP(0.5)), 2 (denoted as NiCoP(2)), and 5 (denoted as NiCoP(5)) °C/min, respectively. Obviously, the NiCoP(2) exhibits excellent electrocatalytic ability toward H_2_O_2_ electroreduction. According to the above optimization results, the mass ratio of 2:5 and the heating rate of 2 °C/min during the phosphorization process are considered optimum to ensure excellent catalytic activity in enhancing electrochemical H_2_O_2_ sensing.

In order to further achieve optimal electrochemical sensing performance of NiCoP NSs electrode materials for H_2_O_2_ electroreduction, the applied voltage and drop-casting amount of NiCoP NSs on GCE was performed to evaluate the electrocatalytic activities of H_2_O_2_ electroreduction using chronoamperometry. Figure 6a shows the current-time curves of NiCoP/GCE at different applied voltages from −0.3 V to −0.6 V with the addition of 5 mM H_2_O_2_. The current responses increase with increasing applied voltage from −0.3 to −0.55 V, and beyond the applied voltage of −0.55 V, the decrease of the current responses is observed. Figure 6b shows the current-time curves of NiCoP/GCE at different drop-casting amounts of NiCoP NSs from 1 to 3 mg mL^−1^ on GCE with the addition of 5 mM H_2_O_2_. When the drop-casting amount of NiCoP increases from 1 to 2 mg mL^−1^, the current responses increase, which results in a decrease in current response. This decrease in current response is due to the mass-transfer limitation of excess catalyst loading [33]. Based on the above optimization results, the applied voltage of −0.55 V and drop-casting amount of 2 mg mL^−1^ NiCoP NSs on GCE are chosen for the following H_2_O_2_ electroreduction sensing.

Figure 7 displays the CV curves of NiCoP/GCE in the presence of 2 mM H_2_O_2_ within the scan rates in the range from 10 to 50 mV s^−1^. It can be observed that the reduction peak current increases as the scan rate increases. In addition, the peak potential shifts to a more negative voltage with the increase of scan rate. The inset of Figure 7 shows that the reduction peak current is proportional to the square root of the scan rate ((mV s^−1^)^1/2^). The linear regression equation can be expressed as I (μA) = −11.346 − 1.661υ^1/2^ ((mV s^−1^)^1/2^) (R^2^ = 0.9973), which means that the electrochemical reduction of H_2_O_2_ on NiCoP/GCE is a diffusion-controlled process.

Under the optimized experimental conditions, the electrochemical performance of the NiCoP/GCE for H_2_O_2_ electroreduction was measured with various H_2_O_2_ concentrations (0~4 mM) to estimate the applicability of the fabricated electrochemical sensing. Figure 8a shows the amperometric response of NiCoP/GCE in 0.1M PBS (pH 7) with successive additions of different concentrations of H_2_O_2_ at an applied voltage of −0.55 V with stirring. It can be observed that amperometric response increases with respect to H_2_O_2_ concentration up to 4 mM. The amperometric response in the concentration range between 0 and 4 mM collect to plot the corresponding calibration plot, is illustrated in Figure 8b. The linear regression equation between amperometric response (ΔI) and the concentration values of H_2_O_2_ (Conc.) can be expressed as ΔI (μA) = −0.29–16.02 Conc. (mM). The calibration plot is linear from 50 μM to 4 mM (R^2^ = 0.9997). The sensitivity, limit of detection (LOD) based on 3 S_b_/m, and limit of quantification (LOQ) based on 10 S_b_/m (S_b_ is the standard deviation of the blank signals for *n* = 3, and m is the slope of the calibration plot) are evaluated as 225.7 μA mM^−1^ cm^−2^, 1.190 μM and 3.97 mM, respectively. Table 1 summarizes the comparison of the electrochemical performance characteristics between the present NiCoP/GCE and some previously reported results based on transition-metal-compound-based electrode materials in electrochemical non-enzymatic H_2_O_2_ sensing [5,14,15,17,32,34,35,36,37], presenting that the proposed NiCoP/GCE is comparable with or even better than those of the recently reported transition-metal-compound-based electrode materials for electrochemical non-enzymatic H_2_O_2_ sensing.

For electrochemical non-enzymatic H_2_O_2_ sensing, anti-interference is a significant indicator for evaluating H_2_O_2_ sensing performance in practical applications. In physiological conditions, some common interfering molecules usually coexist with H_2_O_2_, such as uric acid (UA), ascorbic acid (AA), dopamine (DA), glucose (Glu), and NaCl. Hence, interference testing was conducted to reflect the sensing ability to discriminate the interfering species for determining the target molecule. Figure 9 shows that NiCoP/GCE displays the amperometric response of the target molecule (H_2_O_2_) and some possible interfering molecules (UA, AA, DA, Glu, and NaCl) after successively injecting 400 μM H_2_O_2_, 500 μM UA, 500 μM AA, 500 μM DA, 500 μM NaCl, and 400 μM H_2_O_2_ into 0.1M PBS (pH 7) at an applied voltage of −0.55 V. This result indicates that these interfering molecules do not produce any observable amperometric response and exhibit high selectivity for H_2_O_2_ electroreduction in NiCoP/GCE. To further evaluate the stability of the NiCoP/GCE, the repeatability (Figure 10a) and reproducibility (Figure 10b) of the electrode are examined to assess the efficiency of the electrode by recording the peak current response of 5 mM H_2_O_2_ in 0.1 M PBS (pH 7.0). The repeatability of the NiCoP/GCE is examined by seven consecutive measurements. The relative standard deviation (RSD) is 5.08%. The reproducibility is examined utilizing six different electrodes following the same synthetic procedure, and the obtained RSD is 3.55%, demonstrating that this proposed electrode presents acceptable repeatability and excellent reproducibility, and could be applied as a feasible electrode material for electrochemical non-enzymatic H_2_O_2_ sensing.

## 4. Conclusions

In this study, NiCoP nanosheets (NSs) are successfully synthesized using the hydrothermal and high-temperature phosphorization process. The proposed NiCoP NSs offer a desirable active surface area (exposing abundant electrocatalytic active sites) and excellent atomic/electronic configuration, owing to strong interaction between the transition metal and the phosphide atoms. This strong interaction within NiCoP NSs advances the electrochemical performance, which can be attributed to the synergistic interaction between bimetallic Ni/Co and P atoms and their structure/surface morphological characterization. The electrochemical non-enzymatic H_2_O_2_ sensing-based NiCoP/GCE displays excellent electrocatalytic performance towards H_2_O_2_ electroreduction with a linear range of 0.05~4 mM, a sensitivity of 225.7 μA mM^−1^ cm^−2^, and a limit of detection (LOD) of 1.190 μM. Furthermore, the excellent selectivity of NiCoP/GCE demonstrates a promising future for the developed electrochemical non-enzymatic H_2_O_2_ sensing platform in practical applications.

## Figures and Tables

**Figure 1 nanomaterials-13-00066-f001:**
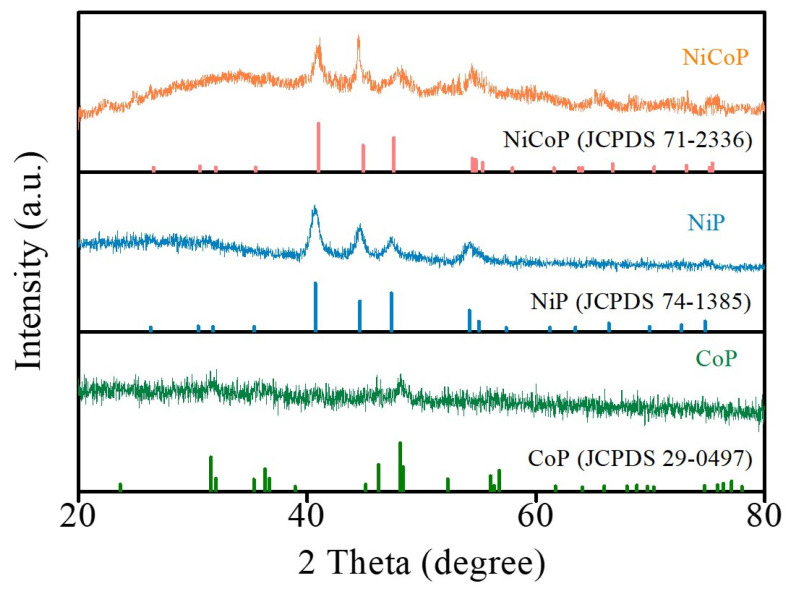
XRD patterns of NiCoP, NiP, and CoP.

**Figure 2 nanomaterials-13-00066-f002:**
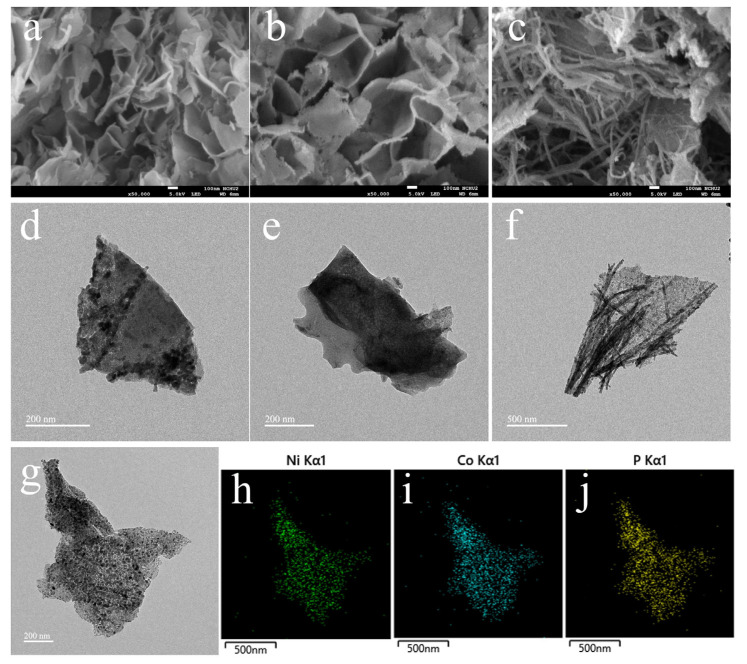
(**a**–**c**) FESEM images and (**d**–**f**) TEM images of NiCoP, NiP, and CoP. (**g**) STEM image of NiCoP and its corresponding elemental mapping images for (**h**) Ni, (**i**) Co, and (**j**) P.

**Figure 3 nanomaterials-13-00066-f003:**
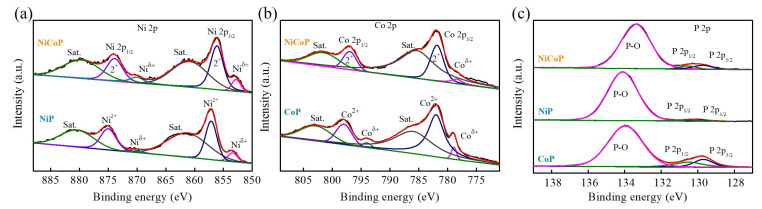
XPS spectra of NiCoP, NiP, and CoP in the (**a**) Ni 2p, (**b**) Co 2p, and (**c**) P 2p regions.

**Figure 4 nanomaterials-13-00066-f004:**
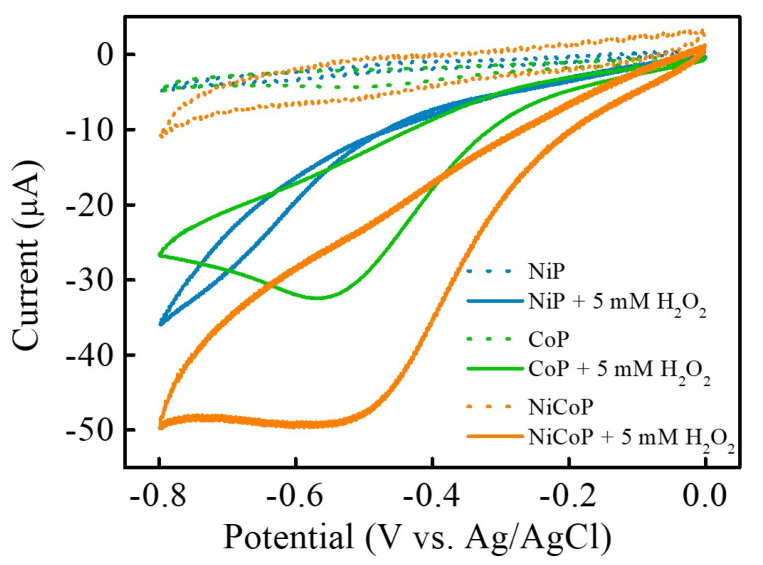
CV curves of NiCoP/GCE (orange line), NiP/GCE (blue line), and CoP/GCE (green line) in 0.1 M PBS (pH 7.0) in the absence (dashed lines) and presence (solid lines) of 5 mM H_2_O_2_ at a scan rate of 50 mV s^−1^.

**Figure 5 nanomaterials-13-00066-f005:**
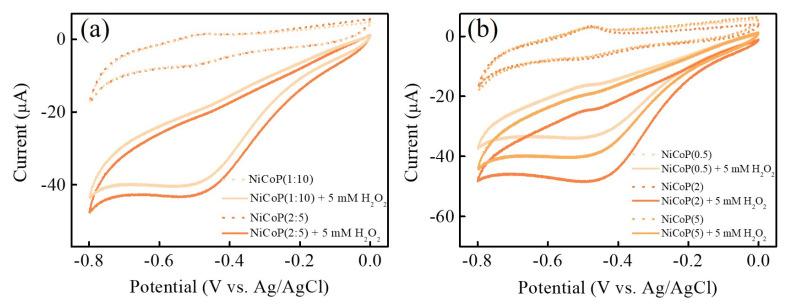
CV curves of NiCoP/GCE prepared with (**a**) different mass ratios (2:5 and 1:10) of NiCo-Pre/NaH_2_PO_2_·H_2_O and (**b**) heating rates (0.5, 2, and 5 °C/min) during the phosphorization process in 0.1 M PBS (pH 7.0) in the absence (dashed lines) and presence (solid lines) of 5 mM H_2_O_2_ at a scan rate of 50 mV s^−1^.

**Figure 6 nanomaterials-13-00066-f006:**
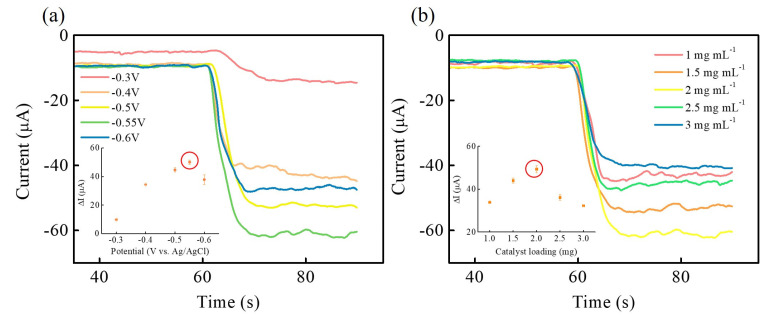
Amperometric responses of NiCoP/GCE at different (**a**) applied voltages: −0.3, −0.4, −0.5, −0.55, and −0.6 V and (**b**) loading amounts: 1, 1.5, 2, 2.5, and 3 mg mL^−1^ with addition of 5 mM H_2_O_2_ in 0.1 M PBS. The optimized performance of NiCoP/GCE marks with red circle in the inset of Figure. (The error bars represent the standard deviation of 3 repeat measurements).

**Figure 7 nanomaterials-13-00066-f007:**
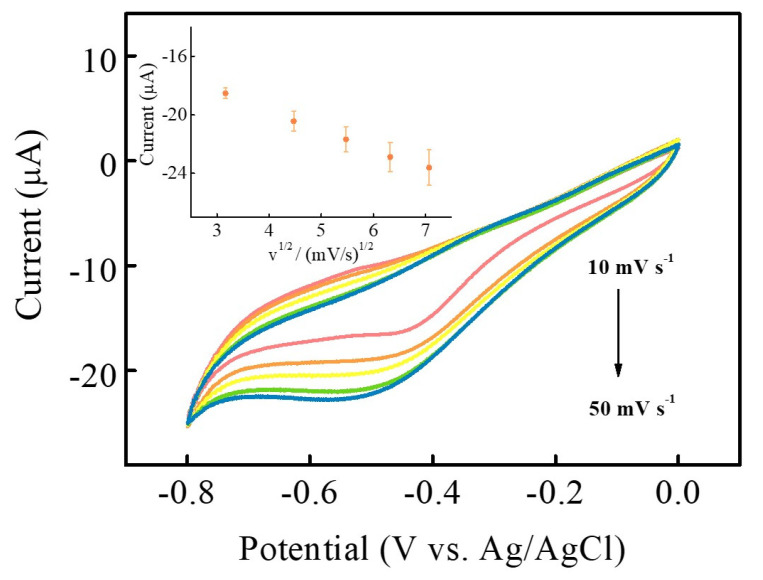
CV curves of NiCoP/GCE in 2 mM H_2_O_2_ at different scan rates from 10 to 50 mV s^−1^. (Inset: the plot of the square root of the scan rate (υ^1/2^) vs. the reduction peak current.) (The error bars represent the standard deviation of 3 repeat measurements).

**Figure 8 nanomaterials-13-00066-f008:**
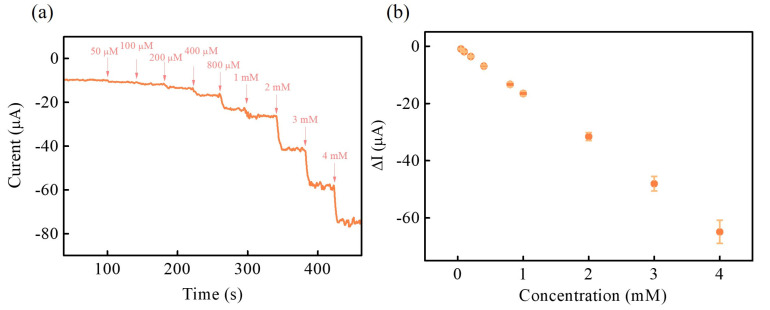
(**a**) Amperometric response of NiCoP/GCE in 0.1 M PBS (pH 7) with successive addition of H_2_O_2_ at applied voltage of −0.55 V, and (**b**) the plot of the amperometric response (ΔI) and the H_2_O_2_ concentration (Conc.).

**Figure 9 nanomaterials-13-00066-f009:**
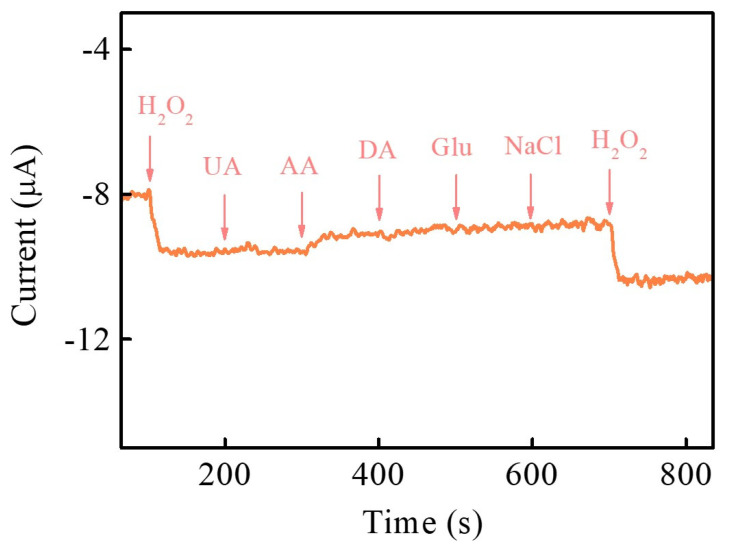
Amperometric response of NiCoP/GCE after successively injecting 400 μM H_2_O_2_, 500 μM UA, 500 μM AA, 500 μM DA, 500 μM NaCl, and 400 μM H_2_O_2_ into 0.1M PBS (pH 7) at applied voltage –0.55 V.

**Figure 10 nanomaterials-13-00066-f010:**
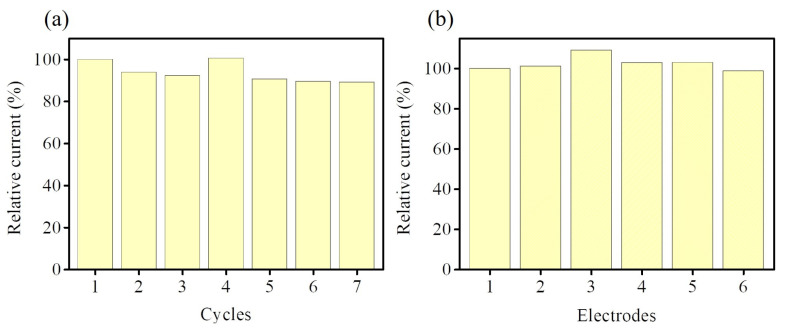
(**a**) Repeatability and (**b**) reproducibility measurements of NiCoP/GCE for the detection of 5 mM H_2_O_2_ in 0.1 M PBS (pH 7.0).

**Table 1 nanomaterials-13-00066-t001:** Performance comparison of electrochemical non-enzymatic H_2_O_2_ sensing based on transition-metal compound electrode materials.

Electrode Materials	Linear Range	Detection Limit (μM)	Sensitivity (μA mM^−1^ cm^−2^)	Reference
Co_3_N	0.002–28 mM	1.000	139.9	[5]
Ni_2_P	0.001–20 mM	0.200	690.7	[14]
Cu_3_P	0.005–0.1 μM	0.002	46181.0	[15]
CoP	0.01–0.1 mM	0.480	3463	[17]
Co_2_P	0.0001–1 mM	0.650	668.6	[32]
Au/CoS_2_	0.022–3.5 mM	20.000	50.0	[34]
NiCo_2_S_4_/rGO	0.025–11.25 mM	0.190	118.5	[35]
Ni_5_P_4_/Ni_2_P	0.05–100 μM	0.020	845.2	[36]
Ni_3_P/Ni_3_S_2_	0.001–20 mM	0.300	134.7	[37]
NiCoP	0.05–4 mM	1.190	225.7	This Work

## Data Availability

Not applicable.

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
