# Peer review of "Synthesis of Bimetallic Ni-Co Phosphide Nanosheets for Electrochemical Non-Enzymatic H_2_O_2_ Sensing"

_nanomaterials, 2022, doi:10.3390/nano13010066_

Round 1

Reviewer 1 Report

Dear Editor,

the manuscript „ Synthesis of bimetallic Ni-Co phosphide nanosheets for electrochemical non-enzymatic H2O2 sensing” (nanomaterials- 2093388) describes Ni-Co phosphide nanosheets as a sensor material for non-enzymatic amperometric detection of hydrogen peroxide. The authors used a number of characterization methods to describe the properties of the sensor material. There are two main problems regarding this manuscript.

First of all, there are already published works on Ni phosphide nanosheets as well as Co phosphide nanosheets, for example: X. Xiong et al., Ni2P nanosheets array as a novel electrochemical catalyst electrode for non-enzymatic H2O2 sensing, 10.1016/j.electacta.2017.09.104; D. Yin et al., Cobalt Phosphide (Co2P) with Notable Electrocatalytic Activity Designed for Sensitive and Selective Enzymeless Bioanalysis of Hydrogen Peroxide, Nanoscale Research Letters volume 16, Article number: 11 (2021), 10.1186/s11671-020-03469-9. These published studies report even better sensor characteristics than the sensor characteristics of the Ni-Co phosphide nanosheets in the proposed manuscript. In particular: “a wide linear range of 0.001-20 mM, a low detection limit of 0.2 μM (S/N = 3), and a high sensitivity of 690.7 μA mM−1 cm−2” //Ni2P nanosheets, 10.1016/j.electacta.2017.09.104//, “response range from 0.001 to 10.0 mM and a low detection limit of 0.65 μM” //Cobalt Phosphide, 10.1186/s11671-020-03469-9//. It is not clear, why the authors got worse characteristics for the monometallic phosphides in their work. Regarding to the above-mentioned articles, the novelty and advantages of bimetallic phosphide nanosheets in the presented manuscript is not clear to justify the publication.

Second, to my opinion, the manuscript is more suitable for the journals dealing with sensors and sensor development, e.g. Sensors. If, however, the editor decide to publish the manuscript in Nanomaterials, minor revision is required before accepting the manuscript for publication.

1)    What is the nature of the NiCo precursor, section 2.2, lines 97-104? According to its preparation, it can be expected that this precursor contains C- and N-atoms because urea is present in the hydrothermal synthesis of the NiCo precursor. Please, provide some analytical results to clarify this aspect.

2)      Fig. 2, please, show EDX spectrum, which include areas of C and N elements.

3)      The authors should include the characteristics of the above mentioned sensors based on Ni- and Co-phosphides /10.1016/j.electacta.2017.09.104, 10.1186/s11671-020-03469-9/ to Table 1 (performance comparison).

4)    Operational stability of the sensor should be evaluated.

Reviewer 2 Report

Chang and Tsai et al. described the manuscript precisely well and developed NiCoP nanosheets (NSs) followed by utilization for non-enzymatic electrochemical H2O2 sensing. They thoroughly investigated the photophysical/chemical properties of NiCoP NSs using SEM, TEM, XRD and XPS studies. The electrochemical strategy for H2O2 sensing would be an innovative idea for future endeavor. Thus, I recommend it's publication in Nanomaterials.

Reviewer 3 Report

This manuscript is dedicated to the synthesis of bimetallic Ni-Co phosphide nanosheets for electrochemical non-enzymatic H2O2 sensing. The authors employ the hydrothermal and high temperature phosphorization fabrication process, while for characterization they use field emission scanning electron microscopy (SEM), field emission transmission electron microscope (FETEM), X-ray diffraction (XRD), and X-ray photoelectron spectroscopy (XPS). The electrochemical properties of NiCoP nanosheets were obtained by cyclic voltammetry (CV) and chronoamperometry (CA), all in all, a much adequate and well-planned strategy for the present task.

Also, the authors discuss, systematize, and clarify the important aspects for such a nanosheet-based material system. More broadly the authors provided valuable ideas for the synthesis of possibly a wider diversity of similar 2D nanosheet-based configurations in the context of of a practical enough technique. The discussion provided is adequate for the present purpose. The well-detailed and at the same time comparative context of the experimental/characterization results clarifies convincingly the authors’ conclusions.

From practical point of view, the reported results thus bring new knowledge and certainly represent an original contribution in the present context.

The authors chose an adequate structure of the manuscript – an excellent point of departure for such a study. Also, they provided a balanced realistic and nicely illustrated presentation of their results and corresponding analysis that is of much scientific and practical interest and adds new knowledge to the field.

The present manuscript is a significant contribution, this work once published would be instructive and suggestive in terms of further studies and to a wider readership.

There are some relatively minor issues with this already excellent manuscript that will need to be addressed before the manuscript becoming suitable for publication, i.e., it can be considered for publication after a minor revision:

1: Authors are a little bit too telegraphic in what concerns the NiCoP nanosheets strong advantage, namely, high accessibility of active sites and rapid carriers transfer, missing especially in the abstract. It will publicize their work better if they are more detailed/literal concerning this issue (in the abstract).

2: The interplay between thermal stability/thermal stress, potential defects in 2D nanosheets and the temperature inherent to the hydro-thermal and high-temperature phosphorization process is relevant to the present results/achievement but it is not mentioned/discussed, this aspect should be also addressed.

3: In the introduction, the authors miss that similar 2D nanosheets with similar complexity and with similar interest concerning the morphology and/or electrochemistry properties to those achieved in the present work have been addressed and guided by theoretical methods of simulation. Examples in which such DFT based theoretical works help understanding synergies between morphological, electrical, and other issues, and even help addressing experimental challenges include: Applied Surface Science 548 (2021) 149275, and Carbon 81(2015) 620-628.

4: In the context of the present discussion of fast charge transfer (page 4 and 5 of the manuscript), are the authors aware of theoretical approaches to address and compare charge transfer between Ni/Co and P atoms in transition metal phosphides? Such data may corroborate the present results.

5: Spell-check and stylistic revision of the paper are still necessary. Some long sentences, misspellings, etc., still are noticeable throughout the text.

Round 2

Reviewer 1 Report

the manuscript can be published inpresent form.